# Diet Intervention Study through Telemedicine Assistance for Systemic Nickel Allergy Syndrome Patients during the COVID-19 Pandemic

**DOI:** 10.3390/nu13082897

**Published:** 2021-08-23

**Authors:** Eleonora Nucera, Angela Rizzi, Raffaella Chini, Sara Giangrossi, Franziska Michaela Lohmeyer, Giuseppe Parrinello, Tania Musca, Giacinto Abele Donato Miggiano, Antonio Gasbarrini, Riccardo Inchingolo

**Affiliations:** 1UOSD Allergologia e Immunologia Clinica, Dipartimento Scienze Mediche e Chirurgiche, Fondazione Policlinico Universitario A. Gemelli IRCCS, 00168 Roma, Italy; eleonora.nucera@policlinicogemelli.it (E.N.); angela.rizzi@policlinicogemelli.it (A.R.); raffaella.chini01@gmail.com (R.C.); p.giuseppe@email.it (G.P.); 2Medicina e Chirurgia Traslazionale, Università Cattolica del Sacro Cuore, 00168 Roma, Italy; gmiggiano@libero.it (G.A.D.M.); antonio.gasbarrini@unicatt.it (A.G.); 3UOC Nutrizione Clinica, Dipartimento Scienze Mediche e Chirurgiche, Fondazione Policlinico Universitario A. Gemelli IRCCS, 00168 Roma, Italy; sara.giangrossi@virgilio.it (S.G.); tania.musca@unicatt.it (T.M.); 4Direzione Scientifica, Fondazione Policlinico Universitario A. Gemelli IRCCS, 00168 Roma, Italy; franziskamichaela.lohmeyer@policlinicogemelli.it; 5UOC Gastroenterologia, Dipartimento Scienze Mediche e Chirurgiche, Fondazione Policlinico Universitario A. Gemelli IRCCS, 00168 Roma, Italy; 6UOC Pneumologia, Dipartimento Scienze Mediche e Chirurgiche, Fondazione Policlinico Universitario A. Gemelli IRCCS, 00168 Roma, Italy

**Keywords:** telemedicine, telenutrition, COVID-19, food allergy, systemic nickel allergy syndrome, balanced diet, nickel, quality of life

## Abstract

Background: Restrictions due to the COVID-19 pandemic limited patients’ access to hospital care. The aims of this study were to assess dietary nutritional status, quality of life (QoL), and adherence to dietary therapy before and after 30-day personalized diet therapy through telenutrition tools in patients with systemic nickel allergic syndrome (SNAS). Methods: Each SNAS patient underwent the following allergological procedures: (a) face-to-face visit (nutritional visit and QoL evaluation) with prescription of one out of five personalized and balanced dietary plans different for calorie intake, (b) video call visit for dietary evaluation and assessment of adherence to diet after 15 days, and (c) video call visit for dietary and QoL evaluation and assessment of adherence to diet therapy after 30 days (end of study). Results: We enrolled 20 SNAS patients. After 15 and 30 days, we found a statistically significant improvement in anthropometric findings after diet therapy, a significant adherence rate to low-nickel diet (60% and 80%, respectively), and an improvement in QoL with an increase in almost all psychometric indices. Conclusions: Our study demonstrates that telenutrition can be a valid tool to monitor nutritional status and adherence to balanced low-Ni diet positively affecting QoL in SNAS patients during the COVID-19 pandemic.

## 1. Introduction

Nickel (Ni), an omnipresent metal and a nutritionally important element, commonly causes allergic contact dermatitis (ACD), affecting nearly 8–19% of adults and 8–10% of children and adolescents in Europe, with a relevant female predominance [1].

A subset of Ni-ACD patients (about 20%) also develops a more severe form of disease called systemic nickel allergy syndrome (SNAS) [2,3], with cutaneous and extra-cutaneous involvement (primarily of the gastrointestinal tract), especially after consumption of Ni-rich plant foods [4], and with cutaneous manifestations even in sites without direct contact [5,6].

Many studies have demonstrated that a significant percentage of SNAS patients improve when placed on a low-Ni diet [4,7,8,9], an effective diagnostic tool and an important first-line non-pharmacological therapeutic measure.

However, it is extremely difficult to follow a restrictive diet and it is socially discriminating. In fact, a low-Ni diet is relatively low in fiber, essential elements, and vitamins, with potentially significant repercussions on the nutritional status, and therefore on the patient’s lifestyle. Furthermore, it is difficult to follow for long periods, especially for those patients who are simultaneously affected by other gastrointestinal comorbidities such as celiac disease or lactose intolerance, all conditions requiring the dietary exclusion of other basic foods (such as milk and its derivatives, dairy products, and vegetables) from the usual dietary regime. All of this can lead to a strong negative impact on patients’ quality of life (QoL) [10,11].

The first example of a balanced diet was proposed by Braga et al. [4]. Researchers developed a specific diet designed to reduce daily Ni-intake under 50 μg/day, within a caloric intake of 2000 kcal/day for men and 1700 kcal/day for women [4]. This approach has moved beyond the classic concept of an elimination diet based on avoiding a predetermined list of forbidden foods.

Recently, QoL of the world’s population has been heavily impacted by the coronavirus disease 2019 (COVID-19) pandemic [12,13].

COVID-19, a contagious disease caused by severe acute respiratory syndrome coronavirus 2 (SARS-CoV-2), was first identified in Wuhan, China, in December 2019. The subsequent spread global disease led to the ongoing pandemic [14]. SARS-CoV-2 infection is characterized by heterogeneity of symptoms. In fact, the infection can be asymptomatic, generate moderate-severe symptoms (interstitial pneumonia, dyspnoea, hypoxia), determine critical symptoms (respiratory failure, shock, or multi-organ dysfunction) [15,16,17].

From the beginning of the pandemic, Italy, similar to other countries in the world, has implemented preventive measures, which include physical or social distancing, and quarantining, in addition to general hygiene recommendations such as ventilation of indoor spaces, covering coughs and sneezes, hand washing, and keeping unwashed hands away from the face. Social distancing (also known as physical distancing) was introduced with the aim to slow the spread of the disease by minimising close contact between individuals, and many patients also limited their accesses to hospital care and visits of physicians [18,19].

In this context, “telemedicine” can have a fundamental role because it allows continuing visits to monitor patients remotely. The World Health Organization (WHO) defines “telemedicine” as “the delivery of health care services, where distance is a critical factor, by all health care professionals using information and communication technologies for the exchange of valid information for diagnosis, treatment and prevention of disease and injuries, research and evaluation, and for the continuing education of health care providers, all in the interests of advancing the health of individuals and their communities” [20].

In the dietary nutritional field, telemedicine specifically includes “tele-nutrition”. The Academy of Nutrition and Dietetics defines “tele-nutrition” as the interactive use of electronic information and telecommunication technologies to implement the Nutrition Care Process (determination, diagnosis, intervention/plan of care, monitoring and nutritional evaluation) carried out remotely [21]. Recently, a step-by-step approach to launch an “electronic nutrition clinic” was proposed for the first time by Farid [22].

Telenutrition could be a crucial tool to remotely monitor and support SNAS patients subjected to restricted and repetitive diets.

The aims of this study were (a) to investigate dietary nutritional status (primary outcome) and, (b) to assess QoL and adherence to dietary therapy (secondary outcomes) of SNAS patients before and after personalized diet therapy through telenutrition tools at the time of the COVID-19 pandemic.

## 2. Materials and Methods

### 2.1. Study Design

We performed a prospective, single-centre, observational study enrolling SNAS patients in order to investigate their dietary nutritional status and QoL, before and after 30-day personalized diet therapy, and adherence to low-nickel diet, through telenutrition tools, during the COVID-19 pandemic.

### 2.2. Setting

The study was performed at the Allergology Unit and Dietetics and Clinical Nutrition Unit of the Fondazione Policlinico Universitario A. Gemelli IRCCS in Rome, Italy. The study was conducted and described according to the STROBE checklist. The Ethics Committee of the Fondazione Policlinico Universitario A. Gemelli IRCCS in Rome, Italy, approved the “Diet Intervention Study through Telemedicine Assistance on SNAS Patients during the COVID-19 PandEmic-DISTANCE Study” (ID 3448; Prot. N. 0040443/20, NCT04894331). All enrolled patients signed an informed consent form for participation in the study.

### 2.3. Participants

We enrolled patients with (a) history of SNAS (coexistence of typical cutaneous and gastrointestinal symptoms); (b) positive nickel-patch test; (c) clinical improvement of at least 70% from baseline after 4 weeks of low-Ni diet excluding all foods with a high content of Ni (Ni 100 µg/kg–Ni > 500 µg/kg), following BraMa-nickel-diet [3,4]; (d) positivity of a double-blind placebo-controlled oral nickel challenge (DBPCO).

Exclusion criteria included (a) age <18 years and >65 years; (b) other organic gastrointestinal diseases, such as peptic ulcer, inflammatory bowel diseases, celiac disease, gastrointestinal infections, and small intestinal bacterial overgrowth; (c) diabetes mellitus; (d) hepatic, renal or cardiac dysfunction; (e) thyroid disease or tumor; (f) concomitant treatment with steroids and/or antihistamines in the previous 4 weeks; (g) pregnancy and lactation; (h) smoking, abuse of alcohol, coffee, tea, and cola intake, and (i) inability to give written informed consent.

From 8 October to 30 November 2020, consecutively enrolled patients underwent a pre-diet face-to-face visit (time 0, T0), completed a 30-day personalized diet, and then underwent a follow-up remote visit through video call (time 2, T2). A further video call was performed 15 days after the start of the diet therapy treatment (time 1, T1).

The study used the commercially available online platform Microsoft Teams, which is included in the Fondazione Policlinico Universitario A. Gemelli IRCCS’s Microsoft Office 365 package.

### 2.4. Variables

We selected two primary outcome measurements (dietary assessment and anthropometric data) to evaluate dietary nutritional status of SNAS patients. Moreover, we adopted a Short-Form 36-Item Health Survey (SF-36v2) questionnaire as tool to evaluate QoL (secondary outcome). Finally, adherence to diet therapy (secondary outcome) was investigated through video call at T1 and T2.

### 2.5. Measurement

Study procedures:

T0 face-to-face visit:-Dietary assessment included nutritional history and eating habits collecting daily and weekly nutritional information with subsequent recommendation to change any incorrect eating habits.-Collection of anthropometric data: weight, height, body mass index (BMI), and body circumferences (wrist, arm, waist, and hips) [23].-QoL evaluation. Each patient answered the SF-36v2 (Italian version) questionnaire. It comprises 36-items measuring eight dimensions of general QoL: physical functioning (10 items), role limitation due to physical health problems (4 items), bodily pain (2 items), general health perceptions (5 items), vitality (4 items), social functioning (2 items), role limitations due to emotional problems (3 items), and general mental health (5 items). Question scores were coded, summed up, and transformed to a scale of 0 (worst possible health state measured by the questionnaire) to 100 (best possible health state) [24].

At the end of the visit, each patient received a personalized diet that took into account not only the Ni content of the foods (Table 1), but also nutritional status, body composition, basal metabolic rate, and energy needs. Furthermore, patients were asked to avoid the use of stainless-steel utensils to reduce Ni contamination during cooking.

The nutritionist prescribed one of five dietary plans, different only for energy intake (1400–1600–1800–2000–2100 Kcal/day) depending on personal energy needs. The energy requirement was calculated on basal metabolic rate (BMR) of the patient according to Harris and Benedict’s [25] and Schofield’s [26] formulas. We chose an average between the two results and, subsequently, multiplied this average value by patient’s physical activity level (PAL). All dietary plans included foods with low-nickel content.

T1 video call visit:-Dietary assessment: collection of anthropometric data.-Assessment of adherence to dietary therapy through question “How many days a week did you adhere to the prescribed dietary treatment?” Two possible mutually exclusive answers: (1) <5 days a week, (2) ≥5 days a week.-T2 video call visit:-Dietary evaluation: collection of anthropometric data.-Assessment of adherence to the dietary treatment.-QoL evaluation.

Figure 1 illustrates the flow-chart of the study.

### 2.6. Statistical Analysis and Sample Size

Given the purely observational nature of the study, no a priori hypotheses were formulated; therefore, a formal calculation of the sample size was not performed. A minimum number of 20 patients to be enrolled was defined.

The sociodemographic and clinical characteristics of the studied population were reported as means and standard deviations for continuous variables and as frequencies and percentages for categorical variables. The chi-squared test was used to evaluate the relationship between frequency of physical activity and BMI. A two-tailed paired Student’s *t*-test was used to compare anthropometric findings at T0, T1, and T2 and changes of SF-36 psychometric indices after low-nickel diet (T2 versus T0). A *p* value < 0.01 was considered significant. Statistical analysis was performed using the IBM Statistical Package for the Social Sciences (SPSS) version 16.0.

## 3. Results

### 3.1. Participants

Twenty-five patients affected by SNAS were screened in this study. Five participants who did not complete all planed follow-up visits online were excluded.

Baseline participants’ characteristics are reported in Table 2.

The majority of patients were females (95%) and 41 ± 8 years old. The assessment of each patient revealed that 55% of patients were normal weight (average BMI 23 ± 3 kg/m^2^; BMI ≥ 18.5 kg/m^2^ and ≤24.9 kg/m^2^), 25% overweight (BMI > 25 kg/m^2^), and 20% underweight (BMI < 18.5 kg/m^2^).

When asked about the most common self-reported symptoms, 65% of participants reported gastro-intestinal disturbances, especially swelling and bloating, even if defecation was described as regular in almost half of the study group. Conversely, cutaneous and respiratory complaints were rarely reported. Three (15%) patients described swallowing problems, while no patient reported chewing disorders.

Moreover, Table 2 shows that 60% of the participants were not at all or 1 day a week engaged in physical activity during the COVID-19 pandemic.

The weekly frequency of physical activity was related to BMI values (*p* = 0.0276, chi-squared test, Figure 2). Only normal weight subjects reported regular physical activity.

Finally, regarding lifestyle, a considerable number of participants drank alcohol, but no patient followed a particular diet, such as vegetarian, vegan, or raw food.

### 3.2. Dietary Behaviours

Table 3 presents the eating habits of the study participants during the COVID-19 pandemic.

Results showed that 50% patients ate 4–5 meals a day and 70% of them at the same time; 95% preferred their home as place for lunch on weekdays. Moreover, patients reported rarely eating outside home (1–2 times a week). Concerning water intake, only 10% of participants consumed at least 2 L per day during the pandemic.

The frequency of consumption for particular food products during the COVID-19 pandemic are presented in Figure 3.

Overall, patients reported healthy eating habits in terms of frequent consumption (at least once a day) of vegetables, fruit, and whole-grain foods (bread/pasta/rice) and concomitant low consumption (few times or less than once a week) of desserts and red and processed meat. Eighty-five percent of enrolled patients did not consume milk and dairy products. Likewise, eighty-five percent of patients did not consume legumes.

### 3.3. Impact of Diet on Anthropometric Findings: Telenutrition Data

When examining the impact of diet therapy on anthropometric findings (weight, BMI, circumference of wrist, arm, waist, and hips, and waist and hip circumference ratio) after 15 and 30 days (T1 and T2, respectively), all mean values of indices showed a statistically significant reduction after diet therapy, except for BMI at T2, wrist circumference and waist and hip circumference ratio (Table 4, two-tailed paired Student’s *t*-test: T1 versus T0 and T2 versus T1, respectively).

### 3.4. Adherence to Low-Nickel Diet

We explored the adherence rate to low-nickel diet using a dedicated questionnaire through telenutrition during the COVID-19 pandemic. Already after 15 days of treatment, the adherence rate, defined by at least 5 days a week of treatment with a low-nickel diet, was significant (60% of the sample). The rate increased after 30 days of treatment, reaching 80% (Figure 4).

### 3.5. Impact of Low-Nickel Diet on QoL of Patients with SNAS during the COVID-19 Pandemic

The low-nickel diet determined a QoL improvement already after 30 days of treatment. In fact, a trend of increase in almost all psychometric indices with statistically significant change for the general health scale (average pre-diet: 38 ± 10, average post-diet: 54 ± 8; *p* = 0.0000), vitality scale (average pre-diet: 49 ± 17, average post-diet: 58 ± 5; *p* = 0.0242) and physical component summary (average pre-diet: 48 ± 6, average post-diet: 53 ± 6; *p* = 0.0369, two-tailed paired Student’s *t*-test) was observed (Figure 5).

## 4. Discussion

In this pilot study, we demonstrated that telenutrition could be a valid tool to monitor nutritional status, adherence to balanced low-nickel diet positively affecting QoL in an Italian population of patients with SNAS during the COVID-19 pandemic.

To date, telenutrition and counselling interventions, as part of the health care program, have a strong impact on health status in terms of weight reduction, hemoglobin A1C, blood pressure, and serum lipids [21,27,28]. However, to the best of our knowledge, no previous study focused on telenutrition to evaluate the dietary nutritional status of SNAS patients and literature on telenutrition as tool to provide clinical support during the COVID-19 pandemic is still scarce. Surprisingly, no study focused on the use of e-health tools in the management of patients with food allergy, growingly considered as a public health burden and defined as the “second wave” of the allergy epidemic, following asthma [29,30].

For decades, food elimination has been the cornerstone of food allergy management. Although immune-modulatory treatments are increasingly used, most patients have yet to rule out one or more culprit foods with potential repercussions on nutritional status [31]. This risk is amplified in SNAS patients due to the ubiquitous nature of the metal and the resulting multiple food restrictions in diet regimens [32], which further compromises QoL [11,33].

Our study, carried out on a predominantly female population, as expected from the literature [1,32,34], showed that the weekly frequency of physical activity of the SNAS patients was related to BMI values. Moreover, only normal weight subjects reported regular physical activity.

In 2015, Lusi et al. found a surprisingly high prevalence of Ni sensitization in overweight women with metabolic syndrome. The low-Ni diet, adopted by researchers, significantly reduced BMI [32].

The “stay-at-home” measure, one of the most common preventive measures adopted by many countries as part of the fight against COVID-19, is potentially related to a change in people’s lifestyle, exercise habits, and diet [35].

Our results enrich the growing literature on the relationship between nutrition status and exercise behaviours during the COVID-19 pandemic. In fact, recently, Özden and Parlar Kiliç examined the effect of social isolation on nutrition and exercise behaviours of nursing students and found that nearly half of students gained weight and the majority did not exercise regularly [36]. In the same period, Deschasaux-Tanguy and colleagues explored changes in dietary intake, physical activity, body weight, and food supply in a large French population highlighting reduction of physical activity in 53% of studied cohort [37]. Similarly, an international online survey, launched in April 2020, confirmed the negative effect of the COVID-19 home confinement on physical activity [35].

Furthermore, we observed preserved healthy eating habits in terms of number of meals per day (most patients ate at least four meals a day), taking main meals at the same time of the day (reported by 70% of participants) and frequency of consumed foods. In particular, we observed high consumption (at least once a day) of “healthy foods” as vegetables, fruit and whole-grain foods and concomitant low consumption (few times or less than once a week) of “unhealthy foods” as desserts, red and processed meat. The consumption of milk and dairy products and legumes were extremely rare. Instead, previous studies highlighted a trend towards unhealthy eating habits during the COVID-19 pandemic. In particular, a cross-sectional study in the United Arab Emirates [38], Middle East, and North Africa region by Cheikh et al. showed a progressive distance from the Mediterranean diet [39]. More recently, Radwan et al. investigated the prevalence and determinants of unhealthy behaviour changes during the COVID-19 lockdown among United Arab Emirates. Increased food intake, especially salty and sweet snacks, was the most common unhealthy behaviour reported by 31.8% of participants [40]. In our study, the attitude towards healthy eating behaviours could probably derive also from SNAS patients’ awareness that dietary interventions are crucial in controlling symptoms due to oral intake of high-nickel foods [9].

Regarding the impact of low-nickel diet, we found an improvement in almost all anthropometric indices already after 15 days. Recently, Tárraga López et al. explored the usual dietary pattern prior to confinement and assessed the adherence to the Mediterranean diet in 490 Spanish adults without evidence of clinically relevant changes in body composition [41]. A recent Italian online survey by Maffoni et al. demonstrated a slight but significant increase in BMI during the COVID-19 pandemic and stratification by lifestyle changes revealed a significant variation in BMI: negative lifestyle changes were correlated with increased BMI and positive lifestyle changes with decreased BMI [42]. It is reasonable to assume that the positive impact of our dietary intervention could depend on the significant adherence rate found after only 15 days, reaching 80% after 1 month of diet.

This study has few limitations. First, it was monocentric. Second, the number of patients was limited partially due to highly selective inclusion/exclusion criteria. Finally, the period of the study (30 days) could influence the generalizability of our results.

Despite these limitations, this study found evidence that telenutrition may allow for monitoring of nutritional status, adherence to a balanced low-nickel diet, and positively affecting QoL in SNAS patients during the COVID-19 pandemic. Furthermore, it confirms the positive effect of low-nickel diet on QoL of SNAS patients [11], as demonstrated with improved physical (general health and physical component summary) and mental health (vitality) domain scales.

This pandemic has caused and continues to cause nutritional and behavioral imbalances, undermining both health and longevity of the world population. One of the possible countermeasures of health systems could be the integration of telemedicine tools in the routine management of patients. This “both/and” approach would allow pursuing a personalized medicine-nutrition strategy also in these contexts. To achieve these goals, future larger trials are needed on how to combine telemedicine and face-to-face visits to optimize the management of food allergy and improve health-related outcomes.

## Figures and Tables

**Figure 1 nutrients-13-02897-f001:**
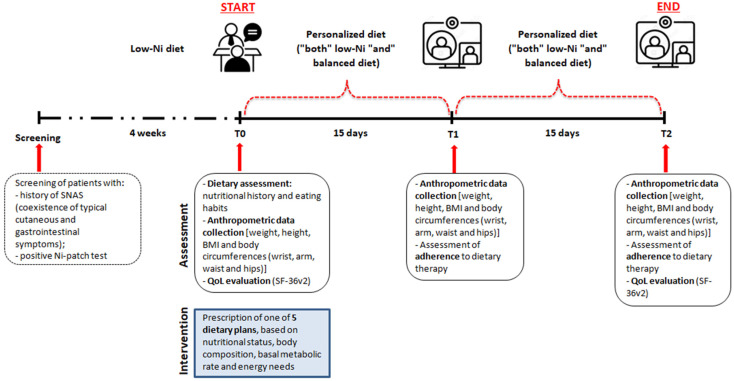
Flow-chart of the study.

**Figure 2 nutrients-13-02897-f002:**
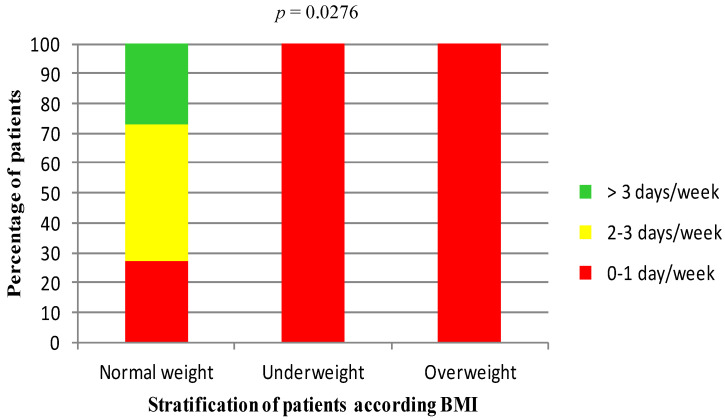
Relationship between frequency of physical activity and BMI. Patients are stratified according to BMI values into three group: normal weight (18.5–24.9 kg/m^2^), underweight (BMI < 18.5 kg/m^2^), and overweight (BMI > 24.9 kg/m^2^). Chi-squared test.

**Figure 3 nutrients-13-02897-f003:**
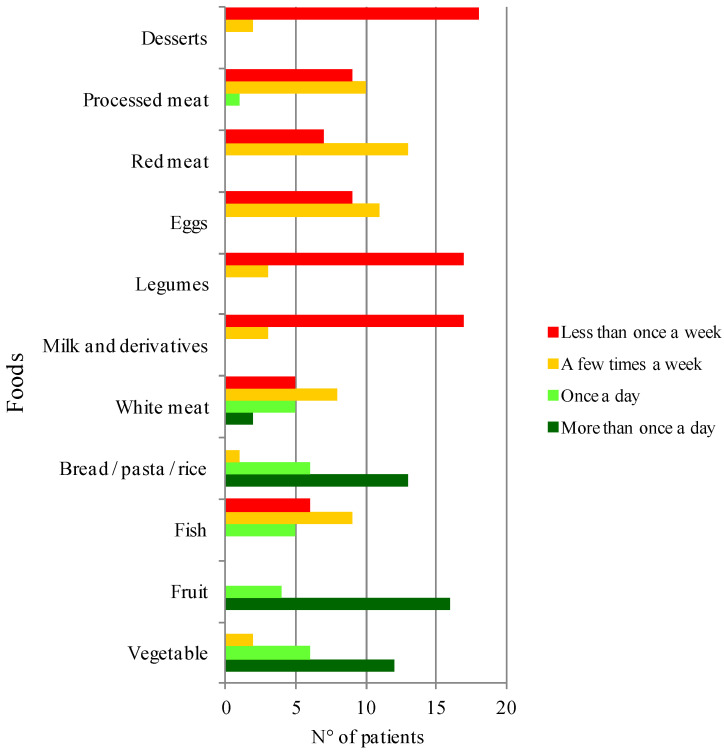
The frequency of consumption of foods during COVID-19 pandemic in the study group of patients with SNAS.

**Figure 4 nutrients-13-02897-f004:**
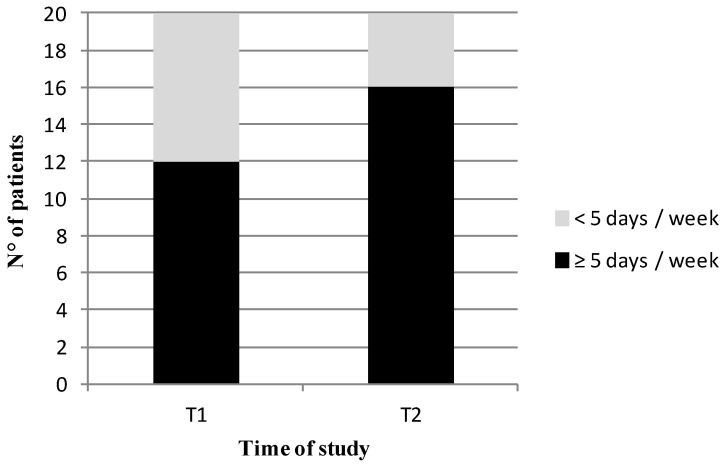
Patients’ adherence to low-nickel diet. Adherence was defined by the number of days per week of low-Nickel diet (≥5 days/week) and was evaluated after 15 days (T1) and at the end of the study (after 30 days, i.e., T2).

**Figure 5 nutrients-13-02897-f005:**
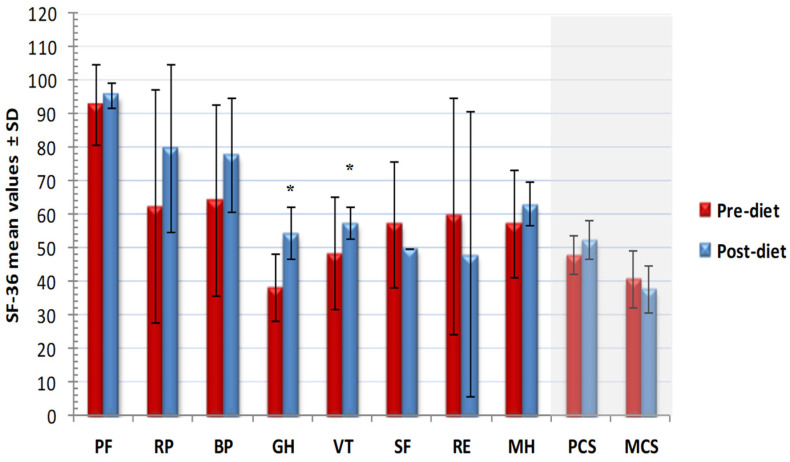
Changes of short form 36 items health survey (SF-36) psychometric indices after low nickel diet. PF, physical function; RP, role-physical; BP, bodily pain; GH, general health; VT, vitality; SF, social function; RE, role-emotional; MH, mental health; PCS, physical component summary; MCS, mental component summary; *: *p* < 0.05 (two-tailed paired Student’s *t*-test).

**Table 1 nutrients-13-02897-t001:** Nickel-rich Foods.

Ni 100 µg/Kg	Ni 200 µg/Kg	Ni 500 µg/Kg	Ni > 500 µg/Kg
Carrots	Apricots	Artichoke	Almonds
Figs	Broccoli	Asparagus	Chickpeas
Lettuce	Corn	Beans	Cocoa
Green Salad	Lobster	Cabbage	Concentrated Tomato
Liquorice	Onions	Cauliflower	Lentils
Mushrooms	Pears	Green Beans	Oats
Plaice and Cod	Raisins	Integral Flour	Peanuts
Rhubarb		Yeast	Walnuts
Tea		Margarine	
		Mussels	
		Oysters	
		Potatoes	
		Peas	
		Plums	
		Spinach	
		Tomatoes	

**Table 2 nutrients-13-02897-t002:** Baseline characteristics of patients suffering from SNAS during COVID-19 pandemic.

	N° of Patients		20
	Sex	male/female	1/19
	Age [years]	mean (±SD)	41 (±8)
Anthropometric data	Weight [Kg]	mean (±SD)	61 (±8)
	Height [cm]	mean (±SD)	165 (±6)
	BMI [Kg/m^2^]	mean (±SD)	23 (±3)
	wrist circumference [cm]	mean (±SD)	15 (±1)
	arm circumference [cm]	mean (±SD)	29 (±4)
	waist circumference [cm]	mean (±SD)	77 (±8)
	hips circumference [cm]	mean (±SD)	99 (±8)
	waist and hips circumference ratio	mean (±SD)	1 (±0)
Clinical findings	Gastro-intestinal symptoms	n (%)	13 (65)
	Cutaneous symptoms	n (%)	2 (10)
	Respiratory symptoms	n (%)	2 (10)
	Defecation: Regular	n (%)	9 (45)
	Constipated	n (%)	7 (35)
	Diarrheal	n (%)	0
	Mixed	n (%)	4 (20)
	Chewing problems	n (%)	0
	Swallowing problems	n (%)	3 (15)
Life-style data	Physical activity:		
	0–1 day/week	n (%)	12 (60)
	2–3 day/week	n (%)	5 (25)
	>3 day/week	n (%)	3 (15)
	Smoking	n (%)	2 (10)
	Alcohol	n (%)	13 (65)
	Particular diet	n (%)	0

**Table 3 nutrients-13-02897-t003:** Dietary behaviours at baseline of SNAS patients during COVID-19 pandemic.

Behaviours	Frequencies/Places/Amounts	N° of Patients	%
N° of meals/day	2–3	6	30
	4–5	10	50
	>5	4	20
Are main meals eaten at the same time?	Yes	14	70
	No	6	30
Lunch on NON-holidays	At home	19	95
	In a company canteen	0	0
	In a school or kinder garden canteen	0	0
	In a restaurant, trattoria, diner	0	0
	In a bar	0	0
	In the workplace	0	0
	At the home pf parents, relatives, friends	1	5
Eat away from home (times/week)	Never	2	10
	1–2	11	55
	2–3	5	25
	>3	2	10
Water	<1/2 L	2	10
	1/2–1 L	11	55
	1–2 L	5	25
	>2 L	2	10

**Table 4 nutrients-13-02897-t004:** Anthropometric findings at baseline, after 15 days and at the end of the study.

		T0 (Baseline)	T1 (after 15 Days)	T2 (after 30 Days)	*p* Value (T1 vs. T0) *	*p* Value (T2 vs. T1) *
weight [Kg]	mean (±SD)	61.2 (±8.1)	60.4 (±7.7)	59.6 (±7.0)	0.0096	0.0116
BMI [kg/m^2^]	mean (±SD)	22.5 (±3.4)	22.0 (±3.0)	221.9 (±2.8)	0.0003	NS
wrist circumference [cm]	mean (±SD)	15.4 (±1.0)	15.4 (±1.1)	15.4 (±1.0)	NS	NS
arm circumference [cm]	mean (±SD)	28.8 (±3.7)	27.7 (±3.0)	27.0 (±2.3)	0.0508	0.0068
waist circumference [cm]	mean (±SD)	77.1 (±8.6)	76.1 (±7.3)	74.5 (±6.4)	0.0375	0.0001
hips circumference [cm]	mean (±SD)	99.3 (±8.2)	98.0 (±7.6)	96.5 (±6.6)	0.0009	0.0012
waist and hip circumference [cm]	mean (±SD)	1 (±0)	1 (±0)	1 (±0)	NS	NS

SD, standard deviation; BMI, body mass index; NS, non-significant. *: The comparison among anthropometric data at baseline (T0), after 15 days and after 30 days of low nickel diet; *p* < 0.05 (two-tailed paired Student’s *t*-test: T1 versus T0 and T2 versus T1, respectively).

## Data Availability

Data sharing is not applicable to this article.

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
