# Peer review of "Diet Intervention Study through Telemedicine Assistance for Systemic Nickel Allergy Syndrome Patients during the COVID-19 Pandemic"

_nutrients, 2021, doi:10.3390/nu13082897_

Round 1
Reviewer 1 Report
The authors describe the effectiveness of tele-medicinal dietary management of nickel-allergic patients with systemic symptoms (SNAS). The paper is written clearly, however, some info is missing.
Major points:
- Include in Intro and Discussion more info on the low-nickel-diet (which food stuff to avoid, where to be careful etc.)
- In the results, the changes in symptoms need to be reported (like in Table 1 "Clinical findings", this would be needed for T0/T1/T2
- Explain clearly (probably in a figure) what was done at which time point, for instance: did these patients have an ongoing low-nickel-diet or were they completely new to the instructions?
Minor points:
- Start Introduction with explanation of SNAS: prevalence, symptoms, dietary management; only then change to Covid-19 measures, then to tele-medicine
- line 53: "caused severe psychological consequences" - delete; include: "patients also limited their accesses to hospital care and visits of physicians".
- line 70: management instead of treatment
- line 72: explain "difficult" and "discriminating" in this context (again, which food, where to be careful)
- line 76-77: explain briefly the corner stones of the low-nickel-diet
- line 77: day instead of die
- line 81: what does mean "before" personalized diet? have these patients been in clinical caretaking before, or did they simply start the diet freshly?
- line 168-170: delete
- line 175: discuss female dominance (known from literature?)
- Table2: what does "Meals at same time" mean - same time as before Covid-19? complete throughout whole manuscript
- line 223: "significant reduction" explain (loss of weight and circumferences, lower BMI - is that good or bad, because you also had some underweight patients before....)
- Discussion: discuss more the topic of the paper (was there a symptom reduction? have patients felt sufficiently secure on their own with the dietary management? did they like tele-medicinal attendance? why is physical activity and BMI important for this paper? more infos again on low-nickel-diet needed: did patients have difficulties in adherence to the diet (even though they were successful)?
- line 257: adherence instead of adherent
- line 275: "physical activity of the SNAS patients"
- line 282: what are nursing students?
- line 290: "same day time for meals as before Covid-19"
- line 306: low-nickel diet: did all these 20 patients have no nickel-diet before?
Author Response
August, 7th 2021
To the Editor and Reviewer
Nutrients
We would like to greatly thank the Reviewer who encouraged the revision of the manuscript.
Reviewers' comments:
Reviewer #1: The authors describe the effectiveness of tele-medicinal dietary management of nickel-allergic patients with systemic symptoms (SNAS). The paper is written clearly, however, some info is missing.
Major points:
Include in Intro and Discussion more info on the low-nickel-diet (which food stuff to avoid, where to be careful etc.)
We thank the Reviewer for the comment. Both sections were extensively modified accordingly.
In the results, the changes in symptoms need to be reported (like in Table 1 "Clinical findings", this would be needed for T0/T1/T2
We thank the Reviewer for the comment that gives us the possibility to better explain the methodology of T1 and T2 video call visits.
As described into “2.5. Measurement” paragraph, dietary (collection of anthropometric data) and adherence assessment (to dietary therapy) were evaluated during T1 video call visit. While, during T2 video call visit, QoL was assessed in addition to what was investigated in T1. The evaluation of symptoms at T1 and T2 was out of purposes of this pilot study.
Explain clearly (probably in a figure) what was done at which time point, for instance: did these patients have an ongoing low-nickel-diet or were they completely new to the instructions?
We thank the Reviewer for the comment. We added Figure 1 in order to illustrate the flow-chart of the study.
Minor points:
Start Introduction with explanation of SNAS: prevalence, symptoms, dietary management; only then change to Covid-19 measures, then to tele-medicine.
We thank the Reviewer for the comment. The Introduction section has been extensively modified accordingly.
line 53: "caused severe psychological consequences" - delete; include: "patients also limited their accesses to hospital care and visits of physicians".
We thank the Reviewer for the comment. The sentence was changed as suggested.
line 70: management instead of treatment
We thank the Reviewer for the comment. The sentence was modified and moved up in the Introduction section.
line 72: explain "difficult" and "discriminating" in this context (again, which food, where to be careful)
line 76-77: explain briefly the corner stones of the low-nickel-diet
We thank the Reviewer for the comment. The Introduction section was modified accordingly.
line 77: day instead of die
We thank the Reviewer for the comment. The word was changed as suggested.
line 81: what does mean "before" personalized diet? have these patients been in clinical caretaking before, or did they simply start the diet freshly?
We thank the Reviewer for the comment that gives us the possibility to better explain the concept of personalized diet.
We enrolled patients with clinical improvement of at least 70% from baseline after 4 weeks of BraMa-Ni-diet [Ref. n° 3-4]. This is an important step into diagnostic approach to patients with suspected SNAS, as described in “2.3. Participants” paragraph.
After the enrollment, each patient received a personalized low-nickel diet, prescribed by Nutritionist, taking into account also the energy requirement and the basal metabolic rate as described in “2.5. Measurement” paragraph.
We modified “2.3. Participants” paragraph accordingly.
line 168-170: delete.
Done.
line 175: discuss female dominance (known from literature?)
We thank the Reviewer for the comment. This aspect of the syndrome was added in Discussion section as required.
Table2: what does "Meals at same time" mean - same time as before Covid-19? complete throughout whole manuscript.
We thank the Reviewer for the comment that gives us the possibility to better explain this question about eating habits. “Meals at same time” means that main meals are eaten at the same time. We modified text and Table 3 accordingly.
line 223: "significant reduction" explain (loss of weight and circumferences, lower BMI - is that good or bad, because you also had some underweight patients before....).
We thank the Reviewer for the comment. We modified accordingly.
Discussion: discuss more the topic of the paper (was there a symptom reduction? have patients felt sufficiently secure on their own with the dietary management? did they like tele-medicinal attendance? why is physical activity and BMI important for this paper? more infos again on low-nickel-diet needed: did patients have difficulties in adherence to the diet (even though they were successful)?
We thank the Reviewer for the comment. We modified accordingly.
line 257: adherence instead of adherent
We thank the Reviewer for the comment. The word was changed as suggested.
line 275: "physical activity of the SNAS patients"
We thank the Reviewer for the comment. The sentence was changed as suggested.
line 282: what are nursing students?
In Özden’s work [Ref. n° 36], the study population consisted of students of the nursing faculty of a university in eastern Turkey.
line 290: "same day time for meals as before Covid-19"
We thank the Reviewer for the comment. The sentence was changed accordingly.
line 306: low-nickel diet: did all these 20 patients have no nickel-diet before?
We thank the Reviewer for the comment. As previously stated, all 20 enrolled patients were instructed to avoid foods with high Ni content as reported in the BraMa diet [Ref. n° 3-4]. After the enrollment, all patients received a balanced diet that took into account not only Ni content of the foods but also nutritional status, body composition, basal metabolic rate and caloric needs of the individual patient. We modified “2.5. Measurement” paragraph accordingly.
With the best regards,
Eleonora Nucera, Angela Rizzi, Raffaella Chini, Sara Giangrossi, Franziska Michaela Lohmeyer, Giuseppe Parrinello, Tania Musca, Giacinto Abele Donato Miggiano, Antonio Gasbarrini and Riccardo Inchingolo
Reviewer 2 Report
Very interesting study and exciting prospect of aiding the SNAS patients. A few recommendations
- Introduction-- It would be helpful to add information on SNAS, epidemiology/pathophysiology/treatments etc for the readers
- Please explain what the low nickel diet entailed, what microgram cut-off did you use, what resources were the patients provided (any apps?)
Author Response
August, 7th 2021
To the Editor and Reviewer
Nutrients
We would like to greatly thank the Reviewer who encouraged the revision of the manuscript.
Reviewer's comments:
Reviewer #2: Very interesting study and exciting prospect of aiding the SNAS patients. A few recommendations.
Introduction-- It would be helpful to add information on SNAS, epidemiology/pathophysiology/treatments etc for the readers
We thank the Reviewer for the comment. Introduction section was extensively modified accordingly.
Please explain what the low nickel diet entailed, what microgram cut-off did you use, what resources were the patients provided (any apps?)
We thank the Reviewer for the comment. We added Table 1 reporting Ni-rich foods. We modified accordingly.
With the best regards,
Eleonora Nucera, Angela Rizzi, Raffaella Chini, Sara Giangrossi, Franziska Michaela Lohmeyer, Giuseppe Parrinello, Tania Musca, Giacinto Abele Donato Miggiano, Antonio Gasbarrini and Riccardo Inchingolo
Round 2
Reviewer 1 Report
All points addressed.
Author Response
August, 19th 2021
To the Reviewer
We would like to greatly thank the Reviewer for the comments.
With the best regards,
Eleonora Nucera, Angela Rizzi, Raffaella Chini, Sara Giangrossi, Franziska Michaela Lohmeyer, Giuseppe Parrinello, Tania Musca, Giacinto Abele Donato Miggiano, Antonio Gasbarrini and Riccardo Inchingolo